# New Solvent-Free Melting-Assisted Preparation of Energetic Compound of Nickel with Imidazole for Combustion Synthesis of Ni-Based Materials

**DOI:** 10.3390/nano11123332

**Published:** 2021-12-08

**Authors:** Oksana V. Komova, Svetlana A. Mukha, Anna M. Ozerova, Olga A. Bulavchenko, Alena A. Pochtar, Arcady V. Ishchenko, Galina V. Odegova, Alexey P. Suknev, Olga V. Netskina

**Affiliations:** Boreskov Institute of Catalysis SB RAS, Pr. Akademika Lavrentieva 5, 630090 Novosibirsk, Russia; msa@catalysis.ru (S.A.M.); ozerova@catalysis.ru (A.M.O.); obulavchenko@catalysis.ru (O.A.B.); po4tar@catalysis.ru (A.A.P.); arcady.ishchenko@gmail.com (A.V.I.); odegova@catalysis.ru (G.V.O.); suknev@catalysis.ru (A.P.S.); netskina@catalysis.ru (O.V.N.)

**Keywords:** nickel complex, imidazole, solvent-free melt synthesis, energetic compound, combustion, Ni, NiO

## Abstract

In this work two approaches to the synthesis of energetic complex compound Ni(Im)_6_(NO_3_)_2_ from imidazole and nicklel (II) nitrate were applied: a traditional synthesis from solution and a solvent-free melting-assisted method. According to infrared spectroscopy, X-ray diffraction, elemental and thermal analysis data, it was shown that the solvent-free melt synthesis is a faster, simpler and environmentally friendly method of Ni(Im)_6_(NO_3_)_2_ preparation. The results show that this compound is a promising precursor for the production of nanocrystalline Ni-NiO materials by air-assisted combustion method. The combustion of this complex together with inorganic supports makes it possible to synthesize supported nickel catalysts for different catalytic processes.

## 1. Introduction

The literature pays great attention to the method of combustion of organometallic precursors for the synthesis of different dispersed materials. It has resulted in dozens of published reviews [1,2,3,4,5] presenting the economy and flexibility of this approach, since varying the conditions makes it possible to synthesize metals and their alloys, simple and complex oxides, supported catalysts, etc.

The solution combustion synthesis with the using glycine-nitrate precursors can be considered as the most studied approach to obtaining highly dispersed NiO, Ni and mixed NiO-Ni composites [6,7,8]. It is well-known that the decomposition of nickel nitrate (without glycine) in air leads to the formation of nickel oxide. As shown in [6], the addition of even a small amount of glycine to it already contributes to the formation of the Ni^0^ impurity. Further studies [9,10,11,12,13] have shown that the predominant formation of the nickel metal phase, as a combustion product, is observed in the fuel-rich conditions. In situ studies have established that Ni^0^ is formed as a result of the reduction of NiO formed at the combustion stage [7,12,14,15]. It was shown that mainly ammonia, as a product of glycine thermolysis, acts as a reducing agent at the post-combustion stage [7,16]. The proposed mechanism of Ni^0^ formation has been confirmed by the thermodynamic calculations [14,15]. A comparative study has shown [17] that the use of organic nitrogen-rich compounds is generally preferred for the production of metal as combustion product. In addition to glycine, organic fuel components such as hexamethylenetetramine [18], hydrazine [19], urea [20] and lecithin [21] are used to obtain nanodispersed metals by the combustion.

The literature shows that the water remaining in the gel precursor after evaporation of the solution of reagents and the conditions of preliminary drying of the precursor have a significant effect on its combustion process and the phase composition of the resulting products [9,13,22,23]. The presence of water in the composition of the precursor leads to a decrease in the temperature of the reaction zone due to heat consumption on evaporation of water and heating of the resulting vapors [22,24,25]. It can be stated that the control over the water content in the composition of the precursor is necessary for the development of reproducible methods for synthesis of nanodispersed materials by the solution combustion method. However, unfortunately, in practice insufficient attention is paid to this important issue.

Synthesis of organometallic salts and complex compounds of nickel with a known structure and composition and their use in the combustion processes allow one to reduce the negative effect of water, which amount is difficult to control in the composition of gel precursors. In this case, the ratio between fuel and oxidizing agent is determined only by the composition of organometallic compounds. Table 1 shows the results on their thermolysis or combustion in order to obtain NiO or Ni samples. For comparison of the oxygen content in the composition of these compounds, the values of the oxygen balance are also indicated. It can be shown that the air-assisted thermolysis (without self-ignition) of the compounds № 1–3 (Table 1) with a high oxygen deficiency (<−130%) led to the formation of nanosized nickel oxide. The combustion process was a characteristic of the compounds № 5–10 (Table 1) with a low oxygen deficiency (>−25%) after thermal action on them. Note that the sulfur contained in the organometallic ligand of the sample № 5 resulted in the formation of nickel sulfide as a combustion product [26].

In our opinion, the use of energetic complex compounds [(Ni^2+^)*_x_*L*_y_*](NO_3_)_2_, where L is a nitrogen-containing CHNO-ligand, is more promising (№ 6–10, Table 1). In such compounds, the fuel and oxidizing structural elements are within the same molecule. The combustion of the complexes № 6–8 was performed in a constant pressure reactor in an inert gas atmosphere. As a rule, this resulted in the formation of porous metallic nickel. On the other hand, for the energetic nitrogen-rich complex compounds № 9 and № 10 (Table 1) the formation of metallic nickel was observed after their air-assisted combustion [27]. This is consistent with the results discussed above that the nitrogen contained in fuel leads to the formation of Ni^0^ even in air. Unfortunately, the effect of nitrogen contained in the structure of complex energetic compounds on the phase composition of combustion or thermolysis products is practically not discussed.

After analysis of literature data, we focused on the lack of information on the use of imidazole (C_3_H_4_N_2_, Im) as a fuel component for the synthesis of materials by combustion. The combustion of the nickel–nitrate–imidazole precursor could be represented as follows:(1)Ni(NO3)2+58 φC3H4N2+52(φ−1)O2 → NiO+ 158 φCO2+54 φH2O+ (8+5φ)8N2

The coefficient φ in this equation is used to characterize the stoichiometry between the fuel (imidazole) and the oxidizing agent (nickel nitrate and oxygen). At φ = 1, the combustion process does not require an additional supply of oxygen; at φ < 1, an excess of oxidant (nitrate anions) is observed in the system; φ > 1 shows an excess of the organic component, i.e., oxygen is required for the complete combustion process.

In part, the lack of interest in this type of fuel for combustion processes is due to its ability to sublimate without a thermolysis stage at temperatures > 100 °C. On the other hand, this feature of imidazole can be easily avoided when it interacts with metal cations to form a complex compound [28,29,30,31].

In this work an energetic complex nitrogen-rich compound Ni(Im)_6_(NO_3_)_2_ (oxygen balance −116%) was prepared and studied. According to Equation (1), the composition corresponds to φ = 935, i.e., fuel-rich conditions. The preparation of this compound was carried out by the traditional method from a solution of nickel nitrate and imidazole [28,29,30,31,37,38], as well as under solvent-free conditions from the melt of these reagents. In the literature, we found only one paper where the synthesis of similar nickel compounds with the composition of NiIm*_n_*X_2_ (*n* = 4, 2, 1; X = Cl, Br, I) was carried out using melting [39]. However, according to studies, the molar ratio of Ni: Im = 1:6 is the most typical for nickel complex compounds [28,29,37]. The importance of development of solvent-free approach is explained by its simplicity, high-speed performance and environmental friendliness. The use of this method eliminates the problem of utilization of wastewater containing heavy metal.

The synthesized samples of Ni(Im)_6_(NO_3_)_2_, the processes and the products of their combustion were investigated by different methods, such as infrared spectroscopy, X-ray diffraction, elemental and thermal analysis and high resolution transmission electron microscopy. In order to adapt the solvent-free approach for the synthesis of supported catalysts, new results on the stability and decomposition of mixture of Ni(Im)_6_(NO_3_)_2_ and **γ**-Al_2_O_3_ were also discussed. Note that **γ**-Al_2_O_3_ is a universal widely used support for catalysts of various processes [40,41,42]. As the results showed, the developed method has proven itself well in the synthesis of NiO/**γ**-Al_2_O_3_ catalyst which was successfully tested in the CO_2_ methanation [43].

## 2. Materials and Methods

The following reagents were used: Ni(NO_3_)_2_·6H_2_O—analytically pure (GOST 4055-70, Reakhim); imidazole C_3_H_4_N_2_—pure grade (TU 6-09-08-1314-78, Vecton); γ-Al_2_O_3_—reagent grade (200 m^2^/g, IK-02-200). For heating, a hot plate IKA C-MAG HS 4 (IKA, Staufen, Germany) was applied.

Synthesis of Ni(Im)_6_(NO_3_)_2_ in the solution. To a solution of imidazole (0.06 mol) in ethanol (10 mL), a solution of Ni(NO_3_)_2_∙6H_2_O (0.01 mol) in ethanol (10 mL) was added dropwise with continuous stirring. The solution turned blue. After the addition of all Ni(NO_3_)_2_∙6H_2_O solution, a violet precipitate was formed. The reaction mixture was stirred for another 15 min, and was left overnight. After that the resulting precipitate was filtered off, washed with a small amount of distilled water and ethanol, evacuated at room temperature, ground in a mortar, and then placed in a desiccator over P_2_O_5_. The yield was 91%.

Synthesis of Ni(Im)_6_(NO_3_)_2_ by the melting. To a melt of imidazole (0.06 mol) in a heat-insulated ceramic crucible, Ni(NO_3_)_2_∙6H_2_O (0.01 mol) was rapidly added with continuous stirring. Temperature of the melt was ~90 °C, setting temperature of the hot plate platform (T_set_) was 170 °C. As a result, a green melt was formed. Then T_set_ was increased to 200 °C, which led to the heating of reaction mixture to 150 °C. The melt quickly turned into a blue lumpy mass, which was stirred (~1 h) until a dry, loose, violet powder was formed. Then the powder was placed in a desiccator over P_2_O_5_, where it cooled to room temperature and stored.

Combustion of Ni(Im)_6_(NO_3_)_2_ synthesized by the melting was carried out in air. A thin layer of Ni(Im)_6_(NO_3_)_2_ complex (2 g) was placed in a high heat-insulated porcelain crucible with a volume of 250 mL, preheated on a hot plate at T_set_ = 500 °C. An intensive gas evolution (thick smoke) was observed during the first few seconds. The resulting product was continued to heat and stir (~15 min) until the glowing sparks on the sample surface completely disappeared and a loose gray-black powder was formed.

Combustion of Ni(Im)_6_(NO_3_)_2_ synthesized by the melting together with γ-Al_2_O_3_ was carried out in air. Alumina with a particle size of <0.04 mm was preliminarily calcined in a muffle furnace at a temperature of 500 °C for 4 h. A powder of the Ni(Im)_6_(NO_3_)_2_ complex synthesized by the melting with a particle size of <0.04 mm was also prepared. Portions (~3 g) of a mechanical mixture of the complex (71.6 wt%) and γ-Al_2_O_3_ (28.4 wt%) were placed in a heat-insulated porcelain crucible with a volume of 500 mL, preheated on a hot plate at T_set_ = 500 °C. At first, a thin layer of green melt was formed, which then quickly turned black and active gas evolution took place, followed by a sharp ignition of the mixture. Heating of the product with periodic stirring was continued for about 2 h, and a gray finely dispersed powder was obtained.

A comparative study of the effect of γ-Al_2_O_3_ on the degree of gasification of the Ni(Im)_6_(NO_3_)_2_ complex synthesized by the melting was performed. In this case small portions (0.025 g) of powders of the Ni(Im)_6_(NO_3_)_2_ complex and a mixture of the complex with γ-Al_2_O_3_ (71.6 wt% Ni(Im)_6_(NO_3_)_2_) were placed in a heat-insulated quartz crucible preheated on a hot plate at T_set_ = 300 °C and heated for 10 min. Then T_set_ was increased to 500 °C and heating was continued for another 15 min. After cooling, the mass of residue was determined.

The content of Ni was determined by inductively coupled plasma atomic emission spectrometry on an Optima 4300 DV instrument (PerkinElmer, Waltham, MA, USA). The contents of C, H, N were determined on an automatic CHNS analyzer EURO EA 3000 (Euro Vector S.p.A., Castellanza, Italy). The samples (0.5–2 mg) were combusted in a vertical reactor in the dynamic regime at 1050 °C in a flow of He with added O_2_.

Attenuated total reflection infrared spectroscopy (ATR FTIR) was performed on an Agilent Cary 600 (Agilent Technologies, Santa Clara, CA, USA) spectrometer equipped with a Gladi ATR (PIKE Technologies, Madison, WI, USA) attachment in the range 250–4000 cm^−1^ without a pretreatment of the samples.

X-ray diffraction analysis (XRD) was performed on a Bruker D8 Advance diffractometer (Bruker AXS GmbH, Karlsruhe, Germany) in the range of angles 5–80° with a step 2θ = 0.05° and the time of accumulation of 5 s in each point using a Lynxeye linear detector. CuK_α_ radiation (λ = 1.5418 Å) was used. Rietveld [44] refinement for quantitative analysis was carried out using the software package Topas V.4.2. The instrumental broadening was described with metallic silicon as reference material. The size of the coherent scattering region (CSR) was calculated using LVol-IB values (i.e., the volume-weighted mean column height based on integral breadth) and LVol-FWHM values (i.e., the volume-weighted mean column height based on full width at half maximum, k = 0.89). The phases were identified using the following data: Ni(Im)_6_(NO_3_)_2_ [PDF 21-1785], Ni [PDF 04-0850], NiO [PDF 47-1049], γ-Al_2_O_3_ [PDF 10-425].

The thermal analysis was performed on a Netzsch STA 449 C Jupiter instrument equipped with a DSC/TG holder (NETZSCH, Selb, Germany) in the temperature range 20–500 °C under a flow of helium and air in corundum crucibles. The heating rate of the samples was 5 °C/min, the weight of the samples was 5 mg.

High-resolution transmission electron microscopy (HRTEM) images were obtained on a JEM-2010 electron microscope (Jeol, Akishima, Japan) with an accelerating voltage of 200 kV and a resolving power of 1.4 Å. The samples to be analyzed were applied to a holey carbon film fixed on a standard copper grid.

## 3. Results and Discussion

### 3.1. Synthesis of Ni(Im)_6_(NO_3_)_2_ Complex Compound

It is known that imidazole is a five-membered heterocycle containing two nitrogen atoms, one is involved in the >N–H group and the other (–N=) has a lone pair of electrons. Due to the last nitrogen atom, imidazole molecule can interact with a metal cation forming M←:N coordination bond and acting as a monodentant ligand. Up to six ligand molecules can surround the central ion forming an octahedral environment [38]. The structure of six-coordinated nickel compound with imidazole—Ni(Im)_6_(NO_3_)_2_—is known. According to XRD data, it has a rhombohedral crystal system with a space group of symmetry R-3 [28].

In our work two approaches were applied for Ni(Im)_6_(NO_3_)_2_ synthesis. In the first case, the synthesis was carried out according to the classical methods [28,29,30,37,38], where interaction of the nickel salt with imidazole occurs in a solvent medium (ethanol in our case) followed by the separation of the resulting complex precipitate and drying until total removal of solvent traces. Note that this approach has a significant drawback—disposal of the used solvent containing heavy metal impurity—and does not meet the principles of green chemistry.

Therefore, in the second case, we used a solvent-free melt method of synthesis. It is known that the melting point of imidazole is ~90 °C [45]. This allowed us to synthesize the complex in the melt at a relatively low temperature in air, which excluded the step of solvent removal and significantly reduced the time of sample drying, since only crystallization water originally present in the nickel hexahydrate should be removed. Thermal analysis data, which will be discussed in detail later, showed that the synthesized complex is stable when heated to 200 °C. Therefore, the heating the reaction medium during synthesis to 150 °C was safe. Water was removed easily during this step, since imidazole is a stronger ligand than water. The presence of crystallization water and its evaporation protect the reaction medium from self-heating. As a result, a dry loose violet powder was formed similar to that prepared via the steps of drying and grinding of the sample precipitated from the solution.

We also noted that the complex formation begins already with intensive grinding of the reagent’s mixture at room temperature, i.e., in the solid phase. This was confirmed by the violet color, which gradually acquires the mixture in a mortar (nickel salt—bright green crystals, imidazole—white powder). However, the resulting product was wet and sticky due to the presence of water. This approach to the Ni(Im)_6_(NO_3_)_2_ synthesis, in comparison with the melting, required further optimization of the process steps in order to increase the product yield and search for optimal drying conditions. The synthesis of Ni(Im)_6_(NO_3_)_2_ from the melt took little time and was more technologically advanced.

According to chemical analysis data, nickel with imidazole in ethanol forms a complex Ni(Im)_6_(NO_3_)_2_. The composition of the sample prepared by the melting method is the same (Table 2). Hence, significant evaporation and sublimation of imidazole does not occur during the synthesis in the melt, which confirms its fast and strong binding with nickel during the complex formation.

In order to confirm the interaction between nickel nitrate and imidazole in the melt, the ATR-FTIR spectra of imidazole and Ni(Im)_6_(NO_3_)_2_ complex compounds prepared in solution and melt were compared (Figure 1). Table 3 shows the assignment of the observed absorption bands in accordance with the published data [29,30]. It is seen that the spectra of nickel–imidazole compounds prepared in solution and melt are identical and differ from the spectrum of imidazole.

It is known that the structure of imidazole is stabilized by strong intermolecular hydrogen bonds (N–H…:N) [46,47]. This leads to rather strong lower-frequency shift of the absorption bands of N–H group stretching vibrations in its infrared spectrum. When imidazole interacts with nickel cations, these hydrogen bonds are destroyed, which leads to higher-frequency shift of ν (N–H) and ν (C–H) absorption bands (Figure 1). Complex formation is also indicated by changes in other spectrum regions (Figure 1, Table 3). The main feature is an appearance of a new absorption band at 260 cm^−1^ for the synthesized compounds, which is absent in imidazole spectrum and should be attributed to the vibrations of Ni–N bond. Note that the spectrum of Ni(Im)_6_(NO_3_)_2_ obtained in our work corresponds to the spectra of this compound described earlier [29,30].

The formation of Ni(Im)_6_(NO_3_)_2_ structure for the complex compounds prepared in solution and melt was also confirmed by XRD analysis. It is obvious from Figure 2 that their XRD patterns are identical and fully correspond to the Ni(Im)_6_(NO_3_)_2_ pattern (PDF 21-1785, space group R-3), which is consistent with the XRD data for this complex [28].

Thus, it was found that the interaction of Ni(NO_3_)_2_∙6H_2_O with six equivalents of imidazole both in solution and melt forms the coordination compound Ni(Im)_6_(NO_3_)_2_. Considering the complete identity of the complexes, the solvent-free melt method may be certainly recommended as the most optimal approach to synthesize such compounds. This complex is stable in air and moisture, and destroys under the action of alkaline or acid solution. Our results demonstrate also its stability in contact with the active surface of γ-Al_2_O_3_ which acid–base properties are described in [41,48]. Figure 3 shows that absorption bands of Ni(Im)_6_(NO_3_)_2_ complex did not change, when it was grinded with γ-Al_2_O_3_ and the grinded mixture was stored for 7 days.

The stability of the Ni(Im)_6_(NO_3_)_2_ complex is explained by the involvement of the N–H bonds in the formation of hydrogen bonds with the oxygen atoms of nitrate anions (N–H…O) during the formation of the crystal lattice. In addition, other types of hydrogen bonds are formed: C–H…N between two neighboring heterocycles and a weaker C–H…O bond between the anion and the ring [28].

### 3.2. Thermal Properties of Ni(Im)_6_(NO_3_)_2_ Prepared by the Melting

The prepared complex nickel compound Ni(Im)_6_(NO_3_)_2_ contains both an organic component and nitrate anions, being a redox system (fuel-oxidizer) capable of exothermic decomposition upon heating [28]. Obviously, the oxygen balance of this complex (−116%) does not allow the organic component to be oxidized only with nitrate-anions in an inert atmosphere. Unfortunately, the published works do not discuss the influence of oxygen on the thermolysis of such energetic compounds. Does oxygen affect the energetics of the step of redox transformations in the imidazoles–nitrates system? In order to understand this issue, we carried out a comparative study of the thermolysis of Ni(Im)_6_(NO_3_)_2_, prepared by the melting method, in helium and air atmosphere. In addition, to develop the synthesis of the supported catalysts via the combustion technique, special attention was paid to the effect of the active γ-Al_2_O_3_ surface [41,48] on the thermolysis of this complex.

Figure 4 shows the thermal analysis data obtained in helium and air atmosphere for the complex compound Ni(Im)_6_(NO_3_)_2_ synthesized by melting as well as for its composition with γ-Al_2_O_3_. It is seen that the initial periods of thermolysis are very similar and practically do not depend on the carrier gas and γ-Al_2_O_3_ presence. In other words, alumina has little effect on the thermal stability of the complex, which is in agreement with the above ATR FTIR data (Figure 3). The investigated complex is stable up to about 200 °C, above this temperature its decomposition begins. The inflection in the TG curve corresponds to a weight loss of 46%. A similar behavior of nickel–imidazole complexes was noted earlier [49,50]. However, as the temperature increases above ~390 °C, the gas composition affects the course of the complex thermolysis. In helium, the slow process of its incomplete decomposition ends with a total weight loss of 64%. In air, the appearance of a new high-temperature step due to the oxidation of the products of incomplete ligands thermolysis is evident. The total weight loss in this case is 90%. Considering the residue weight (10%) at the final temperature, both metallic nickel (calculated residue weight 10%) and its oxide (calculated residue weight 13%) could be formed as a solid decomposition product in air.

When alumina is added to Ni(Im)_6_(NO_3_)_2_, the nature of its decomposition, depending on the gas environment, practically does not change. Since γ-Al_2_O_3_ was calcined at 500 °C, the degree of the complex decomposition in such composition was estimated taking into account the constant support weight during thermolysis. In this case, larger weight loss is observed—70% in helium and 95% in air (Figure 4).

Differentiation of TG curves showed (Figure 5) that the process of Ni(Im)_6_(NO_3_)_2_ complex decomposition is multistep. Three main steps can be distinguished. The first two steps occur in the low-temperature region (200–280 °C) and their characteristics (position, intensity) slightly depend on the gas composition. Small differences are probably associated with the influence of the gas thermal conductivity on the heating rate of the substance crucible (λ_He_ = 156.7 mW/m⋅K, λ_air_ = 26.2 mW/m⋅K). The presence of γ-Al_2_O_3_ has a negligible effect on these steps indicating a high thermal stability of the synthesized complex compound. The third step (350–500 °C) is characteristic only for Ni(Im)_6_(NO_3_)_2_ thermolysis in air, and the maximum temperature (T_max_) for its composition with γ-Al_2_O_3_ is 23 °C lower (Figure 5).

Figure 6 shows a typical correlation of DTG and DSC curves for the decomposition of Ni(Im)_6_(NO_3_)_2_ in air, which includes three steps. As noted above, the first step with a maximum at 232–238 °C is accompanied by the weight loss of about 22.5% due to the elimination of two imidazole molecules. In this case, as expected, a weak endothermic effect is observed [49,50]. According to calculations, the release of two imidazole molecules from this complex increases the oxygen balance from −116% to −95%. As the temperature is further raised, the next step starts reaching a maximum at 252–260 °C and accompanying by the weight loss and an exothermic effect. Since the intensity of this step is practically independent of the gas composition, it can be attributed to redox processes in the imidazoles–nitrates system (complex components). The third high-temperature step (T_max_ = 392–415 °C) occurs only in the presence of oxygen and is associated with exothermic oxidation process of the products of incomplete complex thermolysis. Note again that alumina addition to Ni(Im)_6_(NO_3_)_2_ does not affect its main transformations in steps I and II, but ensures that oxidation step III occurs at lower temperature (T_max_ = 392 °C) (Figure 5).

Based on the thermogravimetric data, the kinetic parameters of Ni(Im)_6_(NO_3_)_2_ thermal decomposition and its composition with γ-Al_2_O_3_ were estimated by the methods of Horowitz–Metzger [51] and Coats–Redfern [52]. In this case, to obtain the activation energy values, one experiment (TG curve) is required, carried out under the sample heating at a constant rate. At the same time, it is assumed that the reaction rate obeys the equation:(2)$$\frac{d\alpha }{dt}=Ae^{-\frac{E}{RT}}{\left(1-\alpha \right)}^{n},$$
where *t*—time, $\frac{d\alpha }{dt}$—reaction rate expressed as a change in substance conversion per unit time, *A*—pre-exponential factor, *E*—activation energy, *R*—molar gas constant, *T*—temperature, *n*—reaction order. For processes performed at a constant heating rate this equation is transformed into:(3)$$\frac{d\alpha }{dT}=\frac{A}{\beta }e^{-\frac{E}{RT}}{\left(1-\alpha \right)}^{n},$$
where *T*—temperature, β—heating rate. It is impossible to solve the integral form of this equation, therefore, in practice, various approximations are applied. Assuming that the reaction corresponds to the first order (*n* = 1), one can use linear approximations of Coats–Redfern [52]
(4)$$\ln\{\frac{-\ln(1-\alpha )}{T^{2}}\}=\ln\frac{AR}{\beta E}(1-\frac{2RT}{E})-\frac{E}{RT}$$
and of Horowitz–Metzger [51]
(5)$$\ln\{\ln(\frac{1}{1-\alpha })\}=\frac{E\theta }{RT_{s}^{2}},$$
where *T_s_*—temperature at which $\left(1-\alpha \right)=\frac{1}{exp}=0.368$, and θ = *T* − *T_s_*.

If several well-defined maxima are observed on the DTG curve, then calculations can be made for each individual peak. Since in the next chapter we study the products of Ni(Im)_6_(NO_3_)_2_ thermolysis in air, the parameters of its individual decomposition as well as in combination with γ-Al_2_O_3_ in oxidizing atmosphere were calculated. Remind our assumption that the first peak (step I) corresponds to the ligands elimination, the second (step II) refers to the redox process between imidazoles and nitrate-anions in the complex compound, and the third (step III) reflects the oxidation of the products of incomplete thermolysis (Figure 4, Figure 5 and Figure 6). The obtained results are summarized in Table 4.

It is shown (Table 4) that all the calculated steps agree well with the kinetics of the first order Equation (4), where *n* = 1. Thus, the use of the Horowitz–Metzger [51] and Coats–Redfern [52] equations is justified. Since the separation of steps I and II on the DTG curves is insufficient, the activation energy values can be considered as estimates. The activation energy of step I calculated by the Horowitz–Metzger model is in the range 154–166 kJ/mol, and in the case of the Coats–Redfern model these values are lower—116–131 kJ/mol. For step II, approximately equal values of 228–243 kJ/mol were calculated by both models. An interesting result was obtained for step III. Table 4 shows that γ-Al_2_O_3_ addition to Ni(Im)_6_(NO_3_)_2_ complex leads to a significant decrease in the activation energy of the exothermic process of oxidation of products of its incomplete decomposition at the stage II. Indeed, for a pure complex it is 422–475 kJ/mol, and for its combination with Al_2_O_3_—268–322 kJ/mol, i.e., the difference is as much as 153–154 kJ/mol for both calculation methods. This explains the obtained results on more complete complex decomposition in the presence of γ-Al_2_O_3_ (Figure 4).

### 3.3. Study of Combustion Products of Ni(Im)_6_(NO_3_)_2_ Complex Prepared by the Melting

Comparative studies of the gasification degree of Ni(Im)_6_(NO_3_)_2_ complex and its mixture with alumina were carried out. The samples combustion was performed under uniform conditions, in air, in a quartz crucible at the platform temperature first T_set_ = 300 °C, then increased to T_set_ = 500 °C and held for 15 min. The obtained data (Table 5) are consistent with the results of thermal analysis. It was also found that in the case of complex composition with γ-Al_2_O_3_, the degree of its gasification increases and reaches the calculated values corresponding to the formation of NiO as the product.

According to XRD data (Figure 7, Table 6), two nanodispersed crystalline phases NiO and Ni^0^ are identified in the product of the complex combustion in air without alumina. We consider that, like at the combustion of glycine–nitrate compositions [7,16], the formation of metallic nickel in air can be associated with the presence of nitrogen in imidazole. As a rule, one of the thermolysis products of N-containing organics is ammonia [16,53,54,55], which can participate in nickel oxide reduction by reaction (2) even in air.

Since the gasification degree of Ni(Im)_6_(NO_3_)_2_ complex is insufficient to form pure NiO and Ni^0^ phases (Table 5), it can be assumed that the products of incomplete complex thermolysis are amorphous. To study their composition, the ATR FTIR spectra were analyzed (Figure 8). Indeed, the spectrum of this sample contains intensive absorption bands. Such a broad poorly structured absorption between 1600 and 1000 cm^−1^ is a characteristic feature of the spectra of both amorphous carbon and amorphous carbon nitride (CN_x_) [56,57,58,59]. In particular, it is generally accepted that the vibrations of −C−C− and −C−N− bonds with C atom in sp^3^ hybridization lies below 1300 cm^−1^, the region 1500–1600 cm^−1^ is associated with the stretching vibrations of double −C=C− and −C=N− bonds with carbon in sp^2^ hybridization for chain structures, and the absorption at 1300–1600 cm^−1^ is due to vibrations in heterocycles [59,60,61]. The range 1300–1600 cm^−1^ is also characteristic of other CN*_x_*H*_y_* compounds such as melem (2,6,10-triamino-s-heptazine C_6_N_7_(NH_2_)_3_), heptazine ((C_6_N_7_)H_3_) and melon ((C_6_N_9_H_3_)_n_) [62]. On the other hand, the band at 2170 cm^−1^ is considered to indicate the formation of the nitrogen-doped carbon phase, since it correlates with the vibrations of −N=C=N− bond with C atom in sp^1^ hybridization [56,60].

A well-known indicator of the carbon nitride (C_3_N_4_) crystalline phase is absorption band at ~810 cm^−1^, which is characteristics of breathing vibration of a triazine ring ((C_6_N_7_)_n_) in g-C_3_N_4_ [63]. The analyzing spectrum contains the band at 813 cm^−1^ but its intensity is low. One can propose that this is due to the low crystallinity degree of carbon nitride. For example, a decrease in the intensity of this band was observed when oxygen enters its structure [62]. The position of low-intensive absorption bands against the background of broad absorption at 1600–1000 cm^−1^ as well as in the low-frequency region allows their assignment to imidazole derivatives. The absorption bands in the high-frequency region at 3600–3100 cm^−1^ are due to the stretching vibrations of the O–H and N–H groups. The CHN-analysis of this sample reveals that the ratio between C:H:N is equal to 1:0.76:0.80. In comparison with imidazole (C_3_H_4_N_2_), there is a decrease in the concentration of hydrogen, but the C/N molar ratio remains the same.

The study of the combustion product of Ni(Im)_6_(NO_3_)_2_ complex by HRTEM actually confirms that the sample mainly consists of NiO particles (Figure 9a,b) with a small fraction of metallic Ni^0^ particles (Figure 9c). Amorphous CN*_x_* phase containing carbon and nitrogen (Figure 9a) and described above by ATR FTIR spectroscopy is also detected.

As was shown by the thermal analysis, the discussed amorphous C*_x_*H*_y_*N*_z_* phase begins to oxidize in air when the temperature reaches ~400 °C (Figure 4, Figure 5 and Figure 6), and the formation of CO, CO_2_ and NH_3_ was detected [43]. Mixing of Ni(Im)_6_(NO_3_)_2_ with γ-Al_2_O_3_ decreases the temperature of this process and provides higher gasification degrees compared with pure complex (Figure 4, Figure 5 and Figure 6, Table 5). Comparison of the ATR FTIR spectra (Figure 8) confirms the low intensity of the absorption bands of amorphous C_x_H_y_N_z_ product in the spectrum of [Ni(Im)_6_(NO_3_)_2_ + γ-Al_2_O_3_] calcined at T_set_ = 500 °C for 15 min. Easy removal of the combustion residue from the surface of the nickel particles (Figure 9a) promotes their more complete oxidation in air. We have used this positive effect of γ-Al_2_O_3_ on the thermolysis of Ni(Im)_6_(NO_3_)_2_ in air to synthesize the supported oxide catalyst 16.4% NiO/γ-Al_2_O_3_ for the methanation of CO_2_ [43]. When preparing this catalyst, we purposely increased the time of its calcination at T_set_ = 500 °C with periodic stirring in air up to 2 h in order to ensure the decomposition of the amorphous C*_x_*H*_y_*N*_z_* phase and oxidation of Ni^0^. XRD confirmed the presence of only γ-Al_2_O_3_ and NiO phases in the catalyst (Table 6, Figure 7). Only XPS showed the presence of Ni^0^ impurity in this catalyst [43], which confirms again the tendency to form metallic nickel upon decomposition of the complex with a high nitrogen content in the ligand even under prolonged oxygen treatment.

## 4. Conclusions

For a long time, the solvent-free combustion method has been considered as a promising scientific and technological approach to the synthesis of multifunctional nanomaterials for various applications. However, little information has been published on the use of energetic complex compounds of nickel in this process. Published data show that such compounds are generally composed of N-containing organic ligands and nitrate anions (Table 1). During their combustion, the formation of metallic nickel was observed not only in an inert atmosphere, but also in air (Table 1). This is in agreement with the results of the well-studied combustion of glycine-nitrate precursors, where it was shown that the reduction of NiO to Ni in air occurs due to ammonia released during the glycine thermolysis [7].

In this work, the energetic complex compound Ni(Im)_6_(NO_3_)_2_ was synthesized by two methods: the traditional one from a solution of nickel nitrate and imidazole and the solvent-free melt approach, which was used for the first time. XRD and ATR FTIR studies have shown that both methods provide the formation of the crystalline compound Ni(Im)_6_(NO_3_)_2_. It can be concluded that the melting method is a convenient, faster and more environmentally friendly method of the synthesis of this compound.

For the complex obtained by the melting method, the process of thermolysis in oxidizing and inert atmospheres was studied, and the values of the activation energy were calculated. It is shown that three main stages were observed during the thermal decomposition of Ni(Im)_6_(NO_3_)_2_. The first (I) endothermic stage (T_max_ = 232–238 °C) corresponded to the release of 2 imidazole molecules from the structure of studied complex compound. The second (II) stage (T_max_ = 252–260 °C) was characterized by an exothermic effect and associated with the beginning of redox interaction in the “imidazole–nitrate” system within the complex structure. The characteristics of the first two stages did not depend on the gas atmosphere. The third (III) stage proceeded only in the presence of oxygen at higher temperatures (T_max_ = 415 °C). It corresponded to the exothermic process of the oxidation of the product of incomplete thermolysis of the complex organic component, which as shown by ATR FTIR spectroscopy and CHN-analysis was an amorphous C_x_H_y_N_z_ phase. According to the XRD, the other combustion products were nanosized phases NiO (80%) and Ni^0^ (20%). We believe that the formation of metallic nickel upon the combustion of Ni(Im)_6_(NO_3_)_2_ in air is explained by a high nitrogen content in imidazole ligand.

It was shown that the contact of the Ni(Im)_6_(NO_3_)_2_ complex with active γ-Al_2_O_3_ surface did not change its thermal stability (stages I and II). However, the presence of alumina led to a decrease in the temperature and activation energy of the air-assisted stage III. According to the literature [64,65,66], this effect may be explained by the ability of the γ-Al_2_O_3_ surface to adsorb and activate the oxygen in air that results in oxidation of amorphous C_x_H_y_N_z_ phase at a lower temperature. The removal of this combustion residue facilitates the oxidation of nickel particles in air. This allowed us to successfully synthesize the supported highly dispersed 16.4% NiO/γ-Al_2_O_3_ catalyst under melt solvent-free conditions. This catalyst demonstrated a high activity in the CO_2_ methanation process [43]. It opens the perspective for synthesis of various supported catalysts.

## Figures and Tables

**Figure 1 nanomaterials-11-03332-f001:**
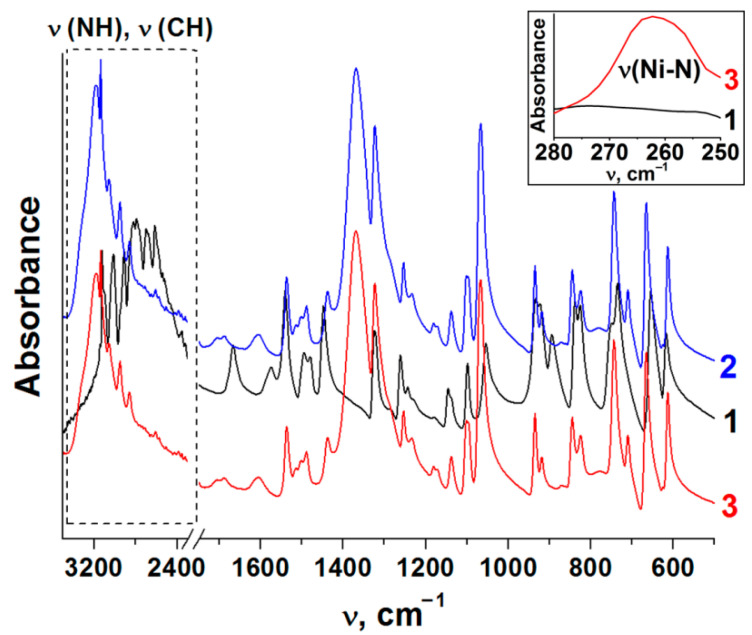
ATR-FTIR spectra of imidazole (1) and Ni(Im)_6_(NO_3_)_2_ complexes prepared in the solution (2) and in the melt (3).

**Figure 2 nanomaterials-11-03332-f002:**
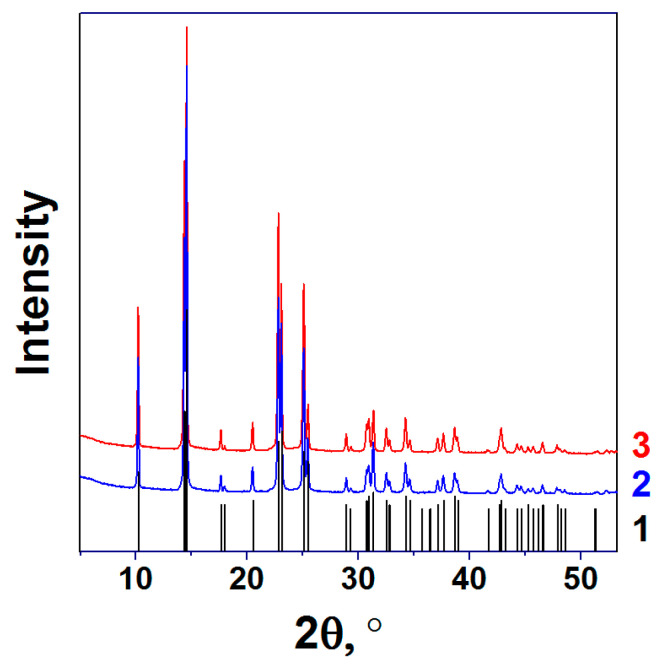
(1) Bar chart of the main XRD peaks for Ni(Im)_6_(NO_3_)_2_ (Nickel imidazole nitrate, space group R-3, PDF 21-1785) and XRD patterns of Ni(Im)_6_(NO_3_)_2_ complexes prepared (2) in the solution and (3) in the melt.

**Figure 3 nanomaterials-11-03332-f003:**
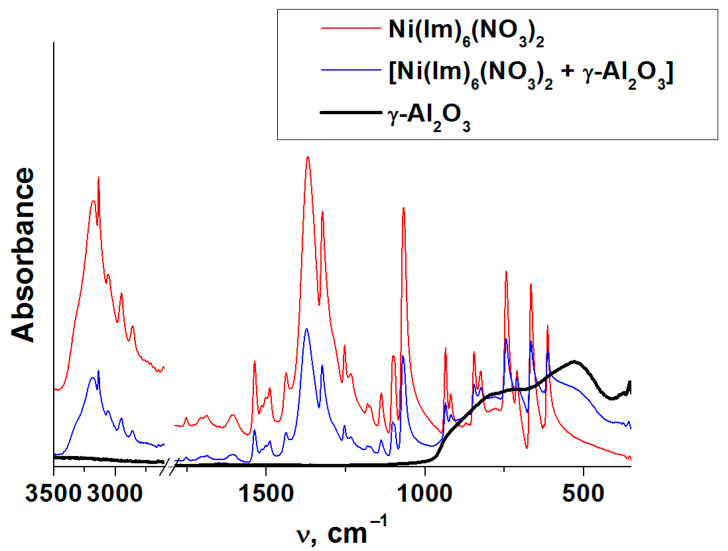
Comparison of the ATR-FTIR spectra of Ni(Im)_6_(NO_3_)_2_ prepared by the melting, the mechanical mixture [Ni(Im)_6_(NO_3_)_2_ + γ-Al_2_O_3_] and γ-Al_2_O_3_.

**Figure 4 nanomaterials-11-03332-f004:**
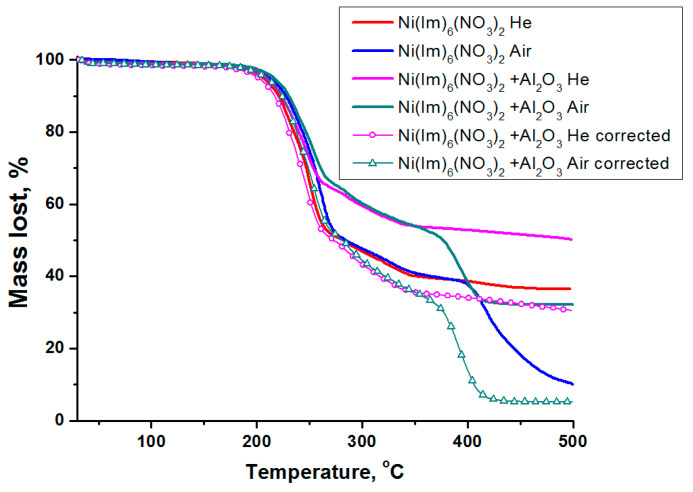
TG curves for thermolysis in He and air of the complex compound Ni(Im)_6_(NO_3_)_2_ prepared by the melting and its composition with γ-Al_2_O_3_. The dots show the recalculated curves for Ni(Im)_6_(NO_3_)_2_ + γ-Al_2_O_3_ composition considering that γ-Al_2_O_3_ weight does not change during the experiment.

**Figure 5 nanomaterials-11-03332-f005:**
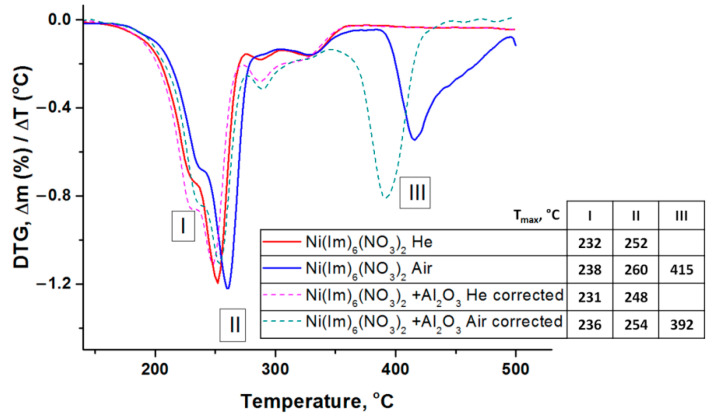
DTG curves for thermolysis in He and air of the complex compound Ni(Im)_6_(NO_3_)_2_ prepared by the melting and corrected DTG curves of its composition with γ-Al_2_O_3_.

**Figure 6 nanomaterials-11-03332-f006:**
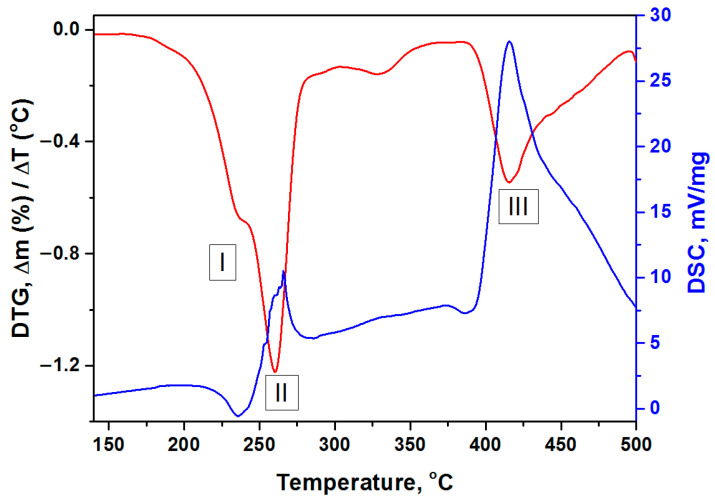
Comparison of DTG and DSC curves for thermolysis in air of the complex compound Ni(Im)_6_(NO_3_)_2_ prepared by the melting.

**Figure 7 nanomaterials-11-03332-f007:**
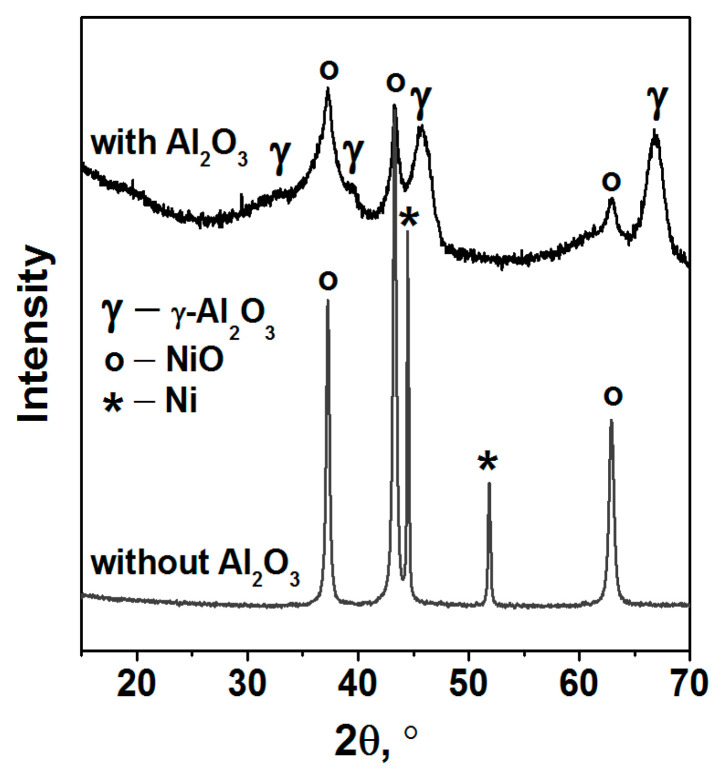
XRD patterns of the thermolysis products of Ni(Im)_6_(NO_3_)_2_ complex prepared by the melting and the mechanical mixture [Ni(Im)_6_(NO_3_)_2_ + γ-Al_2_O_3_].

**Figure 8 nanomaterials-11-03332-f008:**
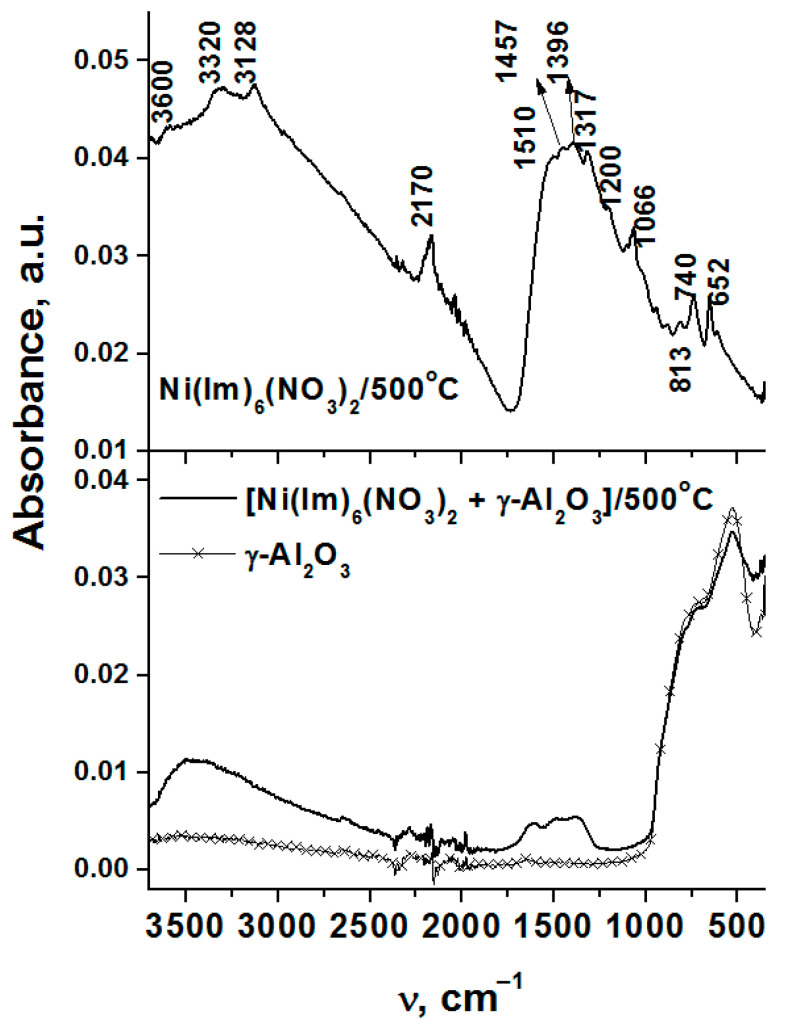
ATR-FTIR spectra of the combustion products of Ni(Im)_6_(NO_3_)_2_ complex prepared by the melting, the mechanical mixture [Ni(Im)_6_(NO_3_)_2_ + γ-Al_2_O_3_] and γ-Al_2_O_3_.

**Figure 9 nanomaterials-11-03332-f009:**
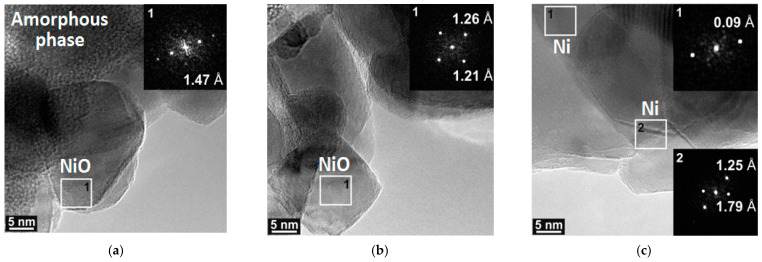
HRTEM images and selected area electron diffraction patterns of the thermolysis products of the Ni(Im)_6_(NO_3_)_2_ complex prepared by the melting. (**a**–**c**) show different locations of the sample.

**Table 1 nanomaterials-11-03332-t001:** The literature data on nickel compounds used for synthesis of NiO or Ni^0^ by thermolysis or combustion processes.

№	Ni Compounds	L = Ligand	Formula	Oxygen Balance	Conditions	Product, Particle Size ^1^	Ref.
1	[Ni(H_2_O)_6_](L)_2_]	d-camphor-10-sulfonate anion (C_10_H_15_O_4_S^−^)	NiC_20_H_30_S_2_O_8_∙6H_2_O	−160%	air, 550 °C, 3 h	NiO 10.81 nm	[32]
2	[Ni(L)_2_(L_1_)H_2_O]	L–acetylacetone (C_5_H_8_O_2_), L_1_–aminopyridine (C_5_H_6_N_2_)	NiC_15_H_22_N_2_O_4_∙H_2_O	−172%	air, 700 °C, 8 h	NiO 40–70 nm	[33]
3	Ni(L)_2_	Octanoate anion (C_8_H_15_OO^−^)	NiC_16_H_30_O_4_	−204%	air, 400–900 °C, 4 h	NiO 24 nm (400 °C) 29 nm (700 °C) 36 nm (800 °C)	[34]
					Ar, 1 atm., 400–800 °C, 4 h	Ni	
4	Ni(L)_2_(NO_3_)_2_·10H_2_O	Hexamethylenetetramine ((CH_2_)_6_N_4_)	NiC_12_H_24_N_10_O_6_∙10H_2_O	−107%	N_2_, 1 atm., 300–700 °C	>500 °C Ni^0^ (10–12 nm) covered by graphite	[35]
5	Ni(L)_2_	Hydrazinesulfinate anion (N_2_H_3_SOO^−^)	NiN_4_H_6_S_2_O_4_	−26%	combustion, air, 500 °C	NiS	[26]
6	[Ni(L)_3_](NO_3_)_2_	Semicarbazide (CH_5_N_3_O)	NiC_3_H_15_N_11_O_9_	−22%	constant pressure reactor, N_2_, 0.1–15 MPa, compacted pellets	Ni	[36]
7	[Ni(L)_2_](NO_3_)_2_	Aminoguanidine (CH_6_N_4_)	NiC_2_H_12_N_10_O_6_	−24%	constant pressure reactor, N_2_, 0.1 MPa, compacted pellets	Ni porous particles 5–15 µm	[36]
					constant pressure reactor, N_2_, 1.5 MPa, compacted pellets	Ni foam	
8	{Ni(L)_1,5_(NO_3_)_2_ ∙ 2H_2_O}_n_ polynuclear complex	Oxalylhydrazide (C_2_H_6_N_4_O_2_)	NiC_3_H_9_N_8_O_9_∙2H_2_O	−11%	constant pressure reactor, N_2_, 0.1 MPa, compacted pellets	Ni aggregates 1 µm	[36]
					constant pressure reactor, N_2_, 1.5 MPa, compacted pellets	Ni melted particles	
9	[Ni(L)_3_](NO_3_)_2_	Semicarbazide (CH_5_N_3_O)	NiC_3_H_15_N_11_O_9_	−22%	constant pressure reactor, air, 0.1 MPa, compacted pellets	Ni with NiO impurity aggregates	[27]
					constant pressure reactor, N_2_, 0.1 MPa, compacted pellets	Ni aggregates	
10	[Ni(L)_2_](NO_3_)_2_	Aminoguanidine (CH_6_N_4_)	NiC_2_H_12_N_10_O_6_	−24%	constant pressure reactor, air, 0.1 MPa, compacted pellets	Ni with NiO impurity 5–15 µm	[27]
					constant pressure reactor, N_2_, 0.1 MPa, compacted pellets	Ni 5–15 µm	

^1^ The values of nm-range correspond to average size of crystallites, the values of μm-range correspond to average size of particles.

**Table 2 nanomaterials-11-03332-t002:** Elemental composition of the complex prepared by different methods.

Method of Synthesis	For Ni(Im)_6_(NO_3_)_2_							
	Calculated, wt%				Found, wt%			
	Ni	C	H	N	Ni	C	H	N
In Solution	9.93	36.57	4.09	33.17	9.30	36.53	4.17	33.66
In Melt					9.57	36.08	4.03	33.45

**Table 3 nanomaterials-11-03332-t003:** Assignment of absorption bands (cm^−1^) in ATR-FTIR spectra of imidazole and Ni(Im)_6_(NO_3_)_2_ complex prepared by the melting and their matching to the published data.

Imidazole This Work	Ni(Im)_6_(NO_3_)_2_ This Work	Ni(Im)_6_(NO_3_)_2_ [30]	Ni(Im)_6_(NO_3_)_2_ [29]	Assignment
3120 (vs) 3104 (sh) 3014 (vs) 2911(vs) 2817 (vs) 2788 (vs) 2696 (vs) 2617 (vs)	3180 (vs) 3135 (vs) 3063 (m) 2950 (m) 2857 (w)	3180(s) 3060(s)	3173 (s) 3135	ν (N–H) ν (C–H)
1540 (s) 1495 (m) 1481 (m) 1448 (s)	1537 (m) 1502 (w) 1490 (w) 1439 (w)	1540 (s) 1500 (m,br)	1540 1503 1490	ν ring
	1370 (vs)	1380 (vs,br)	1373	ν_3_ (NO_3_)
1325 (s)	1324 (s)	1329 (s)	1326	ν ring
1261 (w)	1253 (w)	1256 (s)	1255	δ (CH)
1244 (w)	1233 (w)	1240 (br,sh)	1235	δ (NH)
1146 (w)	1138 (w)	1140 (br)	1141	ν ring
1098 (m)	1098 (m)	1105 (s) 1097 (s)	1104 1099	δ (C–H)
1054 (s)	1067 (s)	1074 (s)	1072	δ (CH)
933 (s)	933 (m)	937 (s)	938	γ (N–H), δ ring, γ (C–H)
921 (s) 895 (m) 838 (s) 825 (s)	921 (w) 845 (m)	920 (m) 848 (s)	921 848	δ ring, γ (C–H)
	825 (w)	828 (m)	827	ν_2_ (NO_3_)
747 (m) 734 (s)	743 (s)	748 (vs)	747	γ (C–H)
	712 (w)	712 (m)	715	ν_4_ (NO_3_)
655 (s) 616 (m)	665 (m) 613 (m)	661 (s)	670 616	γ ring
	260 (w)		261	ν (Ni–N)

vs—very strong, s—strong, m—medium, w—weak, br—broad, sh—shoulder.

**Table 4 nanomaterials-11-03332-t004:** Comparison of the activation energies (E) calculated by the Horowitz–Metzger (HM) and Coats–Redfern (CR) equations for the thermolysis of Ni(Im)_6_(NO_3_)_2_ prepared by the melting.

Model	Parameter	Ni(Im)_6_(NO_3_)_2_			Ni(Im)_6_(NO_3_)_2_ + γ-Al_2_O_3_		
		I	II	III	I	II	III
	T_max_, °C	238	260	415	236	254	392
HM	E, kJ/mol	154	243	475	166	231	322
	n	1	1	1	1	1	1
	α ^1^	0.02–0.15	0.27–0.55	0.68–0.85	0.02–0.18	0.32–0.50	0.73–0.96
	R^2^	0.998	0.996	0.992	0.998	0.992	0.995
CR	E, kJ/mol	116	232	422	131	228	268
	n	1	1	1	1	1	1
	α ^1^	0.02–0.15	0.27–0.55	0.68–0.85	0.02–0.18	0.32–0.50	0.73–0.99
	R^2^	0.997	0.996	0.994	0.998	0.993	0.993

^1^ Sample conversion range calculated basing on current weights related to the beginning and the end of analyzed DTG ranges, and sample weight at the end of thermal analysis at 500 °C.

**Table 5 nanomaterials-11-03332-t005:** Gasification degree at thermal decomposition of Ni(Im)_6_(NO_3_)_2_ complex prepared by the melting and its mixture with γ-Al_2_O_3_.

Sample	Initial Weight of the Sample	Weight of Residue	Gasification Degree	Theoretical Gasification Degree to NiO
Ni(Im)_6_(NO_3_)_2_	0.025 g	0.0103 g	58.8%	87.4%
	0.025 g	0.0095 g	60.8%	
[Ni(Im)_6_(NO_3_)_2_ + Al_2_O_3_]	0.025 g	0.0094 g	87.2%	
	0.025 g	0.0093 g	87.7%	

**Table 6 nanomaterials-11-03332-t006:** XRD analysis data for the thermolysis products of Ni(Im)_6_(NO_3_)_2_ complex prepared by the melting and the mechanical mixture [Ni(Im)_6_(NO_3_)_2_ + γ-Al_2_O_3_]. The thermolysis was carried out in air at T_set_ = 500 °C.

Sample	Phase Composition	Size of Crystallites (nm) LVol-IB/LVol-FWHM
Ni(Im)_6_(NO_3_)_2_/500 °C	80% NiO [PDF 47-1049]	20(1)/26(1)
	20% Ni [PDF 04-0850]	59(5)/74(7)
[Ni(Im)_6_(NO_3_)_2_ + γ-Al_2_O_3_]/500 °C	NiO [PDF 47-1049]	100 ^1^
	γ-Al_2_O_3_ [PDF 10-425]	70 ^1^

^1^ The average crystallite sizes (or average CSR sizes) were calculated by the Scherrer formula using the 200 reflex for NiO and 440 reflex for γ-Al_2_O_3_.

## Data Availability

Data is contained within the article.

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
