# Peer review of "New Solvent-Free Melting-Assisted Preparation of Energetic Compound of Nickel with Imidazole for Combustion Synthesis of Ni-Based Materials"

_nanomaterials, 2021, doi:10.3390/nano11123332_

Round 1

Reviewer 1 Report

This manuscript reports a comparative study on the Ni-complex formed by a solution approach and a solvent-free dry-melt method. The dry-melt derived complex was then ignited to produce a mixture of Ni-NiO. The effects of Al2O3 additives in the complex were also studied, which shows the possibility to achieve NiO loaded Al2O3 catalysts.

To speak frankly, the manuscript is too lengthy, making it diverse from the main topic. The authors are suggested to abbreviate the manuscript, especially the sections of “Introduction” and “Conclusions”.

FESEM images are helpful to show the overlook of the combustion products.

It is also helpful to clarify how the Al2O3 additive hinders the formation of metallic Ni after the combustion.

Several spectra seem to be too similar to each other, for example, Curves 2 and 3 in Figure 1. Please check carefully the original data to be free of any errors.

Author Response

Answer to Reviewer 1

We much appreciate the constructive comments on this manuscript by the Reviewer. All comments were taken into consideration. This allowed us to improve the quality of our manuscript. The changes were marked in yellow.

This manuscript reports a comparative study on the Ni-complex formed by a solution approach and a solvent-free dry-melt method. The dry-melt derived complex was then ignited to produce a mixture of Ni-NiO. The effects of Al2O3 additives in the complex were also studied, which shows the possibility to achieve NiO loaded Al2O3 catalysts.

1 To speak frankly, the manuscript is too lengthy, making it diverse from the main topic. The authors are suggested to abbreviate the manuscript, especially the sections of “Introduction” and “Conclusions”.

Answer: In accordance with comment 1, the Sections of “Introduction” and “Conclusions” have been shortened.

2 FESEM images are helpful to show the overlook of the combustion products.

Answer: In the manuscript, the products of combustion of Ni(Im)6(NO3)2 complex were studied by widely used set of physical methods (XRD, HRTEM, ATR FTIR). Unfortunately, at the moment, due to the deadline, we are not able to apply FESEM method. But we would like to draw the attention of the Reviewer that the new CHN-analysis data for the combustion product have been added in the revised manuscript (Page 16). In comparison with imidazole, it was shown that the C/N molar ratio remains the same, but there is a decrease in the concentration of hydrogen, which, we believe, is explained by the imidazole oxidation and the condensation of the organic residues.

3 It is also helpful to clarify how the Al2O3 additive hinders the formation of metallic Ni after the combustion.

Answer: Our results show that alumina has an influence and reduces the temperature of stage III in the air (Fig.5). This stage corresponds to the oxidation of the product of incomplete thermolysis (amorphous CxHyNz layer) of the studied complex compound. The contact of the complex with g-Al2O3 results in the increase of its gasification degree in the air (Table 5). Easy removal of the combustion residue from the surface of the nickel particles (Fig. 9a) promotes their more complete oxidation in air. According to the literature [64–66], this effect may be explained by the ability of the γ-Al2O3 surface to adsorb and activate the oxygen in the air that results in oxidation of amorphous CxHyNz at a lower temperature.

To meet the reviewer's comment, the comments on the influence of alumina were added on Page 16 and 17 (Conclusions).

4 Several spectra seem to be too similar to each other, for example, Curves 2 and 3 in Figure 1. Please check carefully the original data to be free of any errors.

Answer: In accordance with the comment of the Reviewer, we carefully checked the spectra on Fig.1 and new ATR FTIR spectra were measured on the Agilent Cary 630 spectrometer (see Fig. R1 in the attachment). The results obtained again demonstrate the identity of the spectra of the complex prepared from solution and melt. Note that thorough washing of the sample obtained from the melt with cold alcohol in order to remove traces of the initial reagents also did not change the spectrum of the sample.

Reviewer 2 Report

The work presented for review, entitled "New solvent-free dry-melt preparation method of energetic compound of nickel with imidazole for combustion synthesis of Ni-based materials and catalysts", is suitable for publication in nanomaterials after taking into account some corrections of explanations. I consider the characteristics of the obtained complexes to be valuable in the presented work. The title of the publication, however, is likely to be misleading to the Reader, as it suggests the catalytic properties of the combustion product of the described complexes. The obtained materials have a structure that may or may not have the catalytic capacity, such a claim must be verified and appropriate catalytic tests should be carried out. Is the proposed melt synthesis method safe? The process described in the experimental part appears to be rather risky. Especially when scaling up the process. What is the sensitivity of the obtained energetic compounds materials to the simple stimuli? This is a very important factor in the planned further stages of the process aimed at obtaining a catalyst.   Minor On line 263, replace "и" with "and"  

Author Response

Answer to Reviewer 2

We very much appreciate the reviewer’s remarks and tried to make the appropriate changes in the revised manuscript. The comments have been useful in improving the manuscript.

The work presented for review, entitled "New solvent-free dry-melt preparation method of energetic compound of nickel with imidazole for combustion synthesis of Ni-based materials and catalysts", is suitable for publication in nanomaterials after taking into account some corrections of explanations. I consider the characteristics of the obtained complexes to be valuable in the presented work.

1 The title of the publication, however, is likely to be misleading to the Reader, as it suggests the catalytic properties of the combustion product of the described complexes. The obtained materials have a structure that may or may not have catalytic capacity, such a claim must be verified and appropriate catalytic tests should be carried out.

Answer: Title was corrected as recommended.

2 Is the proposed melt synthesis method safe? The process described in the experimental part appears to be rather risky. Especially when scaling up the process.

Answer: The conditions of the synthesis by melting were studied by us in detail. The optimal ones were selected to ensure the rapid removal of water from the reaction medium and the thermal stability of the synthesized complex compound. Thermal analysis data show that the synthesized complex is stable in air when heated to 200 °C (Fig. 4). Therefore, heating the reaction medium at 150 °C (Tset=200 °C) is safe. An increase in temperature above 200 °C results in thermal destruction of the complex (see Fig. R1 in the attachment).

Taking into account the available values of standard enthalpy of formation (see Table R1 in the attachment) of the compounds and the reaction of complex formation

Ni(NO3)2×6H2O +6C3H4N2 = Ni(C3H4N2)6(NO3)2 + 6H2O­,

the standard enthalpy of this reaction is about 153 kJ/mol (excluding the enthalpy of imidazole fusion ~12 kJ/mol). So when using the nickel hexahydrate, the formation of complex proceeds with the consumption of heat. The reaction becomes exothermic when anhydrous nickel nitrate is used as a starting reagent. In this case, ΔHr0 is about -182 kJ/mol. The safety of the exothermic process is reduced as spontaneous heating of the reaction layer with the decomposition or combustion of the complex is possible.

Table R1. The values of standard enthalpy of formation of compounds [1, 2].

Compound

ΔHf0, kJ/mol

Ni(NO3)2×6H2O

-2211.7

Ni(NO3)2

-425.1

C3H4N2

59.96

Ni(C3H4N2)6(NO3)2

-247*

H2O (g)

-241.9

*-calculated by us from calorimetric measurements of combustion heat using the 6725 Semimicro Calorimeter (Parr Instrument Company, USA).

As a rule, scaling up of laboratory process is a separate complex scientific and technical task, which necessarily includes the modeling stage. The useful properties of the obtaining materials could initiate these studies. At this stage of our research, there is an accumulation of indicators required for scaling.

To meet the reviewer's comment, the information on complex synthesis safety was added on Page 7.

3 What is the sensitivity of the obtained energetic compounds materials to the simple stimuli? This is a very important factor in the planned further stages of the process aimed at obtaining a catalyst.

Answer: The synthesis of Ni(Im)6(NO3)2 from solution has been known for a long time. This compound has been studied and characterized. It is well known that this complex is stable in air and moisture. Our results show also its stability in contact with the active surface of alumina. This complex is destroyed under the action of alkaline solution (0.1 M NaOH) (see Fig. R2 in the attachment). Under these conditions, imidazole is deprotonated and yellow Ni(C3H3N2)2 is partially formed. When exposed to an acid solution (Fig. R2), the sample changes color from violet to blue, which is explained by the competing interaction of a proton with >N-H group of imidazole heterocycle. This leads to the destruction of the Ni-N coordination bond. These changes are clearly visible in dried samples. Also, this sample may be pressed in pellets. In our laboratory practice we used ~1000 kgf pressure.

To meet the reviewer's comment, the information on the sensitivity of the studied compound to the simple stimuli has been added on Pages 9-10.

4 Minor On line 263, replace "и" with "and".

Answer: Thank you very much. It was corrected.

References

1 A. Varma, A.S. Mukasyan, A. S. Rogachev, K. V. Manukyan. Solution Combustion Synthesis of Nanoscale Materials. Chem. Rev. 116 (2016) 14493−14586.

2 M. Zaheeruddin, Z. H. Lodhi. Enthalpies of formation of some cyclic compounds. Phys.Chem. 10 (1991) 111-118. 

Round 2

Reviewer 2 Report

Thanks to the authors for their comprehensive explanations. I recommend the article for publication in its current form.